# A Comparative Study of HA/DBM Compounds Derived from Bovine and Porcine for Bone Regeneration

**DOI:** 10.3390/jfb14090439

**Published:** 2023-08-24

**Authors:** Lina Roldan, Catalina Isaza, Juan Ospina, Carolina Montoya, José Domínguez, Santiago Orrego, Santiago Correa

**Affiliations:** 1Grupo de Investigación en Bioingeniería (GIB), Universidad EAFIT, Medellín 050022, Colombia; lina.roldan@temple.edu (L.R.); cisazaf@eafit.edu.co (C.I.);; 2Department of Oral Health Sciences, Kornberg School of Dentistry, Temple University, Philadelphia, PA 19122, USA; cmontoya@temple.edu (C.M.); sorrego@temple.edu (S.O.); 3Centro de Investigación y Desarrollo Cárnico, Industrias de Alimentos Zenú S.A.S., Grupo Nutresa, Medellín 050044, Colombia; jospinae@zenu.com.co; 4Bioengineering Department, College of Engineering, Temple University, Philadelphia, PA 191122, USA; 5Escuela de Ciencias Aplicadas e Ingeniería, Universidad EAFIT, Medellín 050022, Colombia

**Keywords:** bone graft, demineralized bone matrix (DBM), hydroxyapatite (HA), xenograft

## Abstract

This comparative study investigated the tissue regeneration and inflammatory response induced by xenografts comprised of hydroxyapatite (HA) and demineralized bone matrix (DBM) extracted from porcine (P) and bovine (B) sources. First, extraction of HA and DBM was independently conducted, followed by chemical and morphological characterization. Second, mixtures of HA/DBM were prepared in 50/50 and 60/40 concentrations, and the chemical, morphological, and mechanical properties were evaluated. A rat calvarial defect model was used to evaluate the tissue regeneration and inflammatory responses at 3 and 6 months. The commercial allograft DBM Puros^®^ was used as a clinical reference. Different variables related to tissue regeneration were evaluated, including tissue thickness regeneration (%), amount of regenerated bone area (%), and amount of regenerated collagen area (%). The inflammatory response was evaluated by quantifying the blood vessel area. Overall, tissue regeneration from porcine grafts was superior to bovine. After 3 months of implantation, the tissue thickness regeneration in the 50/50P compound and the commercial DBM was significantly higher (~99%) than in the bovine materials (~23%). The 50/50P and DBM produced higher tissue regeneration than the naturally healed controls. Similar trends were observed for the regenerated bone and collagen areas. The blood vessel area was correlated with tissue regeneration in the first 3 months of evaluation. After 6 months of implantation, HA/DBM compounds showed less regenerated collagen than the DBM-only xenografts. In addition, all animal-derived xenografts improved tissue regeneration compared with the naturally healed defects. No clinical complications associated with any implanted compound were noted.

## 1. Introduction

Bone grafting is the most common surgical method to repair large bone defects [1]. The application of bone graft substitutes in these interventions exhibited a substantial rise, surging from 11.8% in 2008 (10,163 cases) to 23.9% in 2018 (23,838 cases), representing an impressive increase of 134.4% [2]. Reconstructive management of these defects can include autograft, allograft, and animal-derived grafts [3]. Autografts are the gold standard due to their appropriate histocompatibility, osteogenicity, osteoconductivity, and osteoinductivity [4]. However, their use is limited in big-sized defects due to a shortage of supply, donor site morbidity, and the need for a second surgery [5]. Xenografts have been used to circumvent these shortcomings. They have proven to be a suitable alternative to replace hard tissues due to their osteoconductivity, osteoinductivity, and biocompatibility with no supply limitations [6].

Hydroxyapatite (HA) and demineralized bone matrix (DBM) are the most common compounds used as xenografts for bone regeneration [7]. HA is a mineral known for its low cost, biocompatibility, osteoconductivity, and bioactivity [8]. HA has been proven to improve bone regeneration efficiency, making it an excellent candidate for fabricating xenograft bone substitutes [9]. However, HA is limited to small bone defects and is only applied in non-load-bearing regions [10]. Furthermore, HA has a slow resorption rate, adversely affecting bone ingrowth into the scaffold [11]. On the other hand, DBM facilitates the ingrowth of host capillaries, perivascular tissue, and osteoprogenitor cells [12], stimulating blood vessel formation due to the remaining collagen, growth factors, and proteins. DBM is osteoinductive as it contains substances to induce new bone growth [13]. The DBM outcomes in bone regeneration are well documented [14]. However, DBM is not recommended for treating large-sized defects, despite its benefits due to its regenerative inconsistency, mechanical weakness, instability, and handling difficulty [15]. There is a need to create a compound that combines osteoconductivity (acting as “soil”) and osteoinductivity (acting as “fertilizer”) for treating large-sized defects [16]. HA/DBM compounds aim to mimic the structure of natural bone by providing a desirable environment for cell attachment and proliferation. These compounds may offer both osteoconductive and osteoinductive effects [17], and have demonstrated variable results in terms of bone regeneration [18]. A unique advantage of HA/DBM compounds over DBM- or HA-only grafts is that the inflammatory response induced by DBM may foster the resorption of HA, which accelerates bone regeneration [19].

DBM and HA can be extracted from different sources. Bovine, equine, and porcine substitutes are the most well-known bone substitutes [20]. Bovine-derived xenografts are the most widely used graft materials currently dominating the market in maxillofacial surgery and orthopedics [21]. However, these xenografts are associated with clinical complications such as foreign body reactions, encapsulation, and soft tissue fenestration [22]. Porcine-derived xenografts have emerged as a solution to overcome the limitations of bovine sources due to their similarity to the human bone structure [23]. Grafts from various origins and compositions have different bone regeneration potentials due to their physiochemical characteristics [16]. For example, in the case of bovine- and porcine-derived bone grafts, some differences have been attributed to the porosity, crystallinity, pore distribution, and particle size [24]. In addition, the availability of the animal source directly impacts production costs and can trigger ethical concerns. The extraction processes of HA and DBM can also influence the physicochemical properties of the graft and, thus, their regeneration capabilities [24]. Incorrect extraction techniques may leave residual proteins that produce adverse reactions after implantation [25]. To the best of our knowledge, a small number of studies have focused on comparing the bone regeneration performance of xenografts from bovine and porcine sources [26]. However, without standardized extraction procedures, tissue regeneration and inflammation differences cannot be well established [27]. Therefore, comparing the bone regeneration performance of xenografts from different animal sources and extraction processes is difficult. This study compared the regeneration and inflammatory response of bone substitutes fabricated from HA and DBM derived from bovine (B) and porcine (P) sources extracted using a unique protocol. After extraction, the individual HA and DBM were characterized and mixed in proportions of 50/50 and 60/40 to obtain clinically useful compounds. Then, the compounds were implanted into critical-sized calvarial defects in rats. Three and six months after implantation, a histomorphometry study was performed to evaluate the regeneration of bone and the inflammatory response.

## 2. Materials and Methods

### 2.1. Bone Extraction and Preparation

The cortical bone (femur) of adult bovines and porcines of ~3.3 years and nine months old were harvested from a certified slaughterhouse (Industria de Alimentos Zenú S.A.S, Medellin, Colombia). The femur was selected because of its sizeable uniform area. Extracted bones were preserved in ice and stored at −20 °C. After extraction, the femurs were cleaned using a No. 15-blade to remove the cartilage, ligaments, and soft tissues that adhered to the bone. Next, femurs were cut at the metaphysis using an electric saw (ST295AI, Torrey, Querétaro, Mexico). The diaphysis was cut into blocks of 3 cm^2^ and was used for HA and DBM extractions. Finally, the cortical bone blocks were immersed in detergent in water to remove fat residues [28].

### 2.2. HA Extraction

HA extraction was performed as described previously [29] (Figure 1). Briefly, bone blocks were thawed and boiled in water to remove any attached soft tissue and then dried at 100 °C (Memmert100–800, Memmert, Schwabach, Germany) [30]. Defatting was performed by submerging the bones in 70% ethanol and air drying them. Before deproteinization, the bone pieces were crushed and ground using a jaw crusher (MJ 90, MEKA, Ankara, Turkey). The deproteinization was performed in two steps. First, the ground material was soaked in sodium hypochlorite (NaOH at 2%) to eliminate dirt and proteins from the surface. Second, the material was placed in oxalic acid for complete deproteinization, followed by drying at 100 °C. The resulting material was milled again to reduce the particle size [31]. Finally, for the complete removal of organic compounds, the material was calcined at 1100 °C (D8, Terrigeno, Medellín, Colombia).

### 2.3. DBM Extraction

DBM powder was prepared from the cortical portion of the bones (Figure 1). First, to decalcify the tissue, cortical bone pieces (3 cm^2^) were submerged in a 4% formaldehyde solution [32]. Subsequently, the samples were washed with DI water and dried (Memmert 100–800, Memmert, Schwabach, Germany). Next, the bone blocks were crushed and ground in a ball mill (PM100, Retsch, Haan, Germany). Fatty components were removed by soaking in 70% ethanol. Bone was demineralized with HCl and washed with di-water [32]. Finally, the samples were freeze-dried and stored under aseptic conditions until further use.

### 2.4. HA/DBM Compound Preparation

The HA/DBM compound was prepared by mixing the powder of both materials with carboxymethyl cellulose (CMC) (Gelycel, Montevideo, Uruguay) as a polymeric binding material and 1% (*v*/*v*) doxycycline hydrochloride (BioBasic, Markham, ON, Canada). Mixing was performed in a clean room using sterile instruments and a shaker (CimarecSP131325Q, Thermo Scientific, Waltham, MA, USA). The mixed components were stirred until the mix had a “dough” consistency. After mixing, the resulting compound was placed into a sterile syringe. The following concentrations (HA/DBM) were fabricated for each animal source: porcine (P) and bovine (B) at 50/50P, 60/40P, 50/50B, and 60/40B each. No cross-animal mixtures were performed. A summary of the DBM and HA extraction procedure is presented in Figure 1. The sterilization of the compound was performed with low-dose gamma irradiation (~15 KgY).

### 2.5. HA and DBM Individual Characterization

The morphology and physicochemical properties of the extracted HA and DBM were characterized. First, the particle size distribution of the extracted HA was analyzed using a particle size analyzer (Cilas 1064, Cilas, Orleans, France). Pulverized samples (5 g) were dispersed in isopropyl alcohol and sonicated for 60 s before analysis. Second, the morphologies of the extracted HA and DBM were evaluated using scanning electron microscopy (SEM) (ZEISS EVO 10, Zeiss, Oberkochen, Germany) at an accelerating voltage of 20 kV. The extracted materials and compounds were evaluated using energy-dispersive X-ray spectroscopy (EDS). Three randomly selected images were analyzed for each sample. Third, an X-ray diffractometer (Empyrean, Malvern Panalytical, Malvern, UK) using Cu radiation (λ = 1.5403 Å) at 45 kV and 40 mA was used to verify the presence of the crystalline phases in the extracted HA. Diffraction data were collected from 10° ≤ 2θ ≤ 100° with a step size of 0.05° and measurement time of 54 s per 2θ interval. Semi-quantification of the phases was performed using the High Score Plus software (Malvern Panalytical, Malvern, UK) using the Rietveld method and the ICSD FIZ Karlsruhe 2012-1 [22]. Five samples (N = 5) were scanned for each group. Commercial HA (Medical Group, Saint-Priest, France) was used as the control. Fourier transform infrared spectroscopy (FTIR) was used to verify the chemistry of the extracted DBM. The spectra of the DBM samples (N = 3 per group) were recorded using a Spectrum Two FT-IR spectrometer (PerkinElmer, Waltham, MA, USA) equipped with a single reflection diamond for attenuated total internal reflection (ATR) operating with a resolution of 4 cm^−1^ and a wavenumber range of 400–4000 cm^−1^. A commercial DBM Puros^®^ (Zimmer Biomet, Warsaw, IN, USA) was also scanned as a clinical reference.

### 2.6. HA/DBM Compound Characterization

The HA/DBM compounds were characterized for their morphology, elemental composition, injection force, rheological properties, biodegradation profile, and their inflammatory and osteoblastic differentiation response. The injection force was performed by dispensing the compounds through 5 cm^3^ syringes assembled with a 27-gauge needle. Before the experiment, the syringes were sonicated for 1 min to remove bubbles. The injection force was measured using a mechanical testing machine (Instron 4411, Instron, Norwood, MA, USA) equipped with a 150 N load cell. The load was applied to the plunger to simulate a flow rate of 2 mL/h [33]. The rheological properties of the compounds were assessed using a Physica MCR 101 controlled stress rheometer (Anton Paar, Graz, Austria) with serrated parallel plates (diameter 25 mm). The rheological properties were measured under linear viscoelastic conditions. A frequency (from 0.1 to 100 Hz) with a fixed strain (0.5%) was performed to measure the storage modulus (G′) and loss modulus (G″) [33]. The in vitro biodegradation profile was assessed following the ASTM F1634-95 standard to ensure consistent and standardized evaluation (Appendix A) [34]. Additional inflammatory and osteoblastic differentiation responses were evaluated with an ELISA test by measuring the expression of TNF-α and IL-6 in bone-marrow-derived mesenchymal stem cells (BMSCs) (Appendix A).

### 2.7. Clinical User Evaluation

User testing was performed by interviewing 12 possible clinical users (four maxillofacial surgeons, two periodontists, and six orthopedists). After a verbal explanation of the purpose of the evaluation, the users were allowed time to manipulate the compounds. Subsequently, the specialists were asked to blindly select the best compound according to its adhesion, malleability, and stability. Finally, descriptive statistical analysis for user testing was performed by calculating the characteristic percentages of preferences.

### 2.8. In-Vivo Evaluation

#### 2.8.1. Animal Model and Surgical Procedures

All animal handling and surgical procedures were performed following the ethical guidelines of the Universidad de Antioquia (Medellin, Colombia) and under the approval record session No. 138 (9 February 2021). Forty-four (N = 44) male adult Wistar Norvegicus Rats (Bioterio, Universidad de Antioquia, Medellin, Colombia), five months old and weighing 500 ± 100 g, were used. The animals were housed under standard conditions, with ad libitum access to food and water. The animals were anesthetized with isoflurane, followed by an intraperitoneal injection of 5% ketamine (100 mg/kg) and 2% xylazine (10 mg/kg) [35]. Following anesthesia, a 1.5 cm sagittal incision was made in the skull skin using a No. 15-blade. Enrofloxacin and gentamicin (5 mg/kg) were administered to the exposed skulls. Two osteotomies of 4 mm (without compromising the dura) were performed on the right and left parietal bone using an NSK Surgical AP handpiece (NSK, Tokio, Japan) and a 4 mm trephine. Saline irrigation was manually applied using a syringe [36]. Then, ~0.2 mL of the HA/DBM compounds (50/50P, 60/40P, 50/50B, and 60/40B) were randomly implanted on the right-side defect (Appendix A). As a clinical reference, the Puros^®^ Cancellous Particulate Allograft was also implanted. The left side defect was used as a control (no material implantation). Postoperatively, the subcutaneous fascia and skin were saturated with Vicryl 5.0. The animals were injected subcutaneously with enrofloxacin 5% (5 mg/kg, every 12 h, for five days) and flunixin meglumine (2.5 mg/kg) with tramadol (2.5 mg/kg, every 12 h for three days) [37]. Appendix A outlines the experimental design. The animals were monitored routinely for 15 days after surgery for weight loss, behavior, healing of the surgery site, and temperature (Appendix A) [38]. After three (N = 22) and six months (N = 22), the animals were euthanized with 20% CO_2_ for 5 min. The period of evaluation at three and six months enabled monitoring the long-term effectiveness of HA/DBM in promoting bone growth and regeneration, as was previously reported [39]. Moreover, it yielded valuable safety information, as any adverse reactions or complications could become evident during this period.

#### 2.8.2. Histological Analysis

Histological sections of the calvarial defects were used to analyze bone healing and inflammation. After euthanasia, the skulls were immersed in 4% paraformaldehyde for 36 h [40]. After fixation, the samples were decalcified in 5% nitric acid solution for 24 h at room temperature and then embedded in paraffin wax [41]. Sagittal sections of 5 μm thickness were stained with hematoxylin−eosin (HE) and examined by light microscopy. All of the images were captured at the same magnification and settings. The images were processed using an Olympus CX43 microscope and CellSens(r) image-processing software (Olympus, Center Valley, PA, USA). An experienced pathologist performed the histological analyses.

Four measurements were conducted to quantify the tissue regeneration and inflammation, including the percentage of calvarial thickness regeneration, percentage of regenerated bone and collagen areas, and percentage of blood vessel area. First, the percentage of calvarial thickness regeneration was measured as the ratio of the average thickness of the regenerated tissue and the average of the original calvarial thickness. The averages were calculated along the calvarial in four equally spaced locations [42]. Second, the percentages of regenerated bone and collagen areas were measured by dividing the area of regenerated tissue (bone or collagen) by the total regenerated area [43]. Finally, the inflammation was quantified by measuring the ratio between the area of blood vessels (mm^2^) and the total regenerated area (mm^2^). This last measurement included hard and soft tissue within the location of the defect [44]. Appendix A contains additional information and specific methods for calculating these variables. Quantitative analysis was performed using ImageJ software (NIH, Bethesda, MD, USA).

### 2.9. Statistical Analysis

Statistical analyses were calculated using GraphPad Prism 9 software (GraphPad Software, Boston, MA, USA). The results are presented as the mean ± SD. After the model assumptions were met, ANOVA with a significance of 0.05 and the Tukey post hoc test were used for multiple comparisons with a 95% confidence level.

## 3. Results

### 3.1. HA and DBM Characterization

The particle size distribution of HA extracted from both animals was similar, ranging from 3 to 500 mm (Figure 2a). In both groups, the highest particle size accumulation was ~25% with ~100 µm. Particles smaller than 45 µm (line red in the figure) were filtered through a sieve and were used for the compound preparation and further characterization; 45 µm particles corresponded to 20% and 18% of the total HA obtained from bovine and porcine sources. The morphology of HA from bovine bone (Figure 2(bI)) showed irregularly shaped particles, whereas porcine particles were slightly regular/round with soft edges (Figure 2(bII)). The DBM’s morphology was comparable between the two species, without evident differences (Figure 2(bIII,bIV)). The elemental analysis of HA confirmed the presence of peaks of calcium and phosphorus with a Ca/P ratio of ~1.68 (Figure 2c). The diffractograms of HA from both sources were characteristic of highly crystalline materials that matched the standard diffractogram of apatite (code: 98-026-1063) (Figure 2d). Characteristic diffraction peaks for apatite were found at 2q = 31.7°, 32.2°, and 32.9°, corresponding to the (211), (112), and (300) planes, respectively [45]. Secondary peaks at 2q = 39.8, 25.9°, and 34° corresponding to the (310), (002), and (202) planes, respectively, were also evident in both materials. Broader peaks were obtained for porcine HA, potentially related to larger crystal sizes, micro stresses, gradients, and chemical heterogeneities [46]. The FTIR spectra of the HA extracted from both animals were comparable to those observed for the commercial material (Figure 2e). All of the samples displayed transmittance bands at 560 cm^−1^, 600 cm^−1^, 1023 cm^−1^, and 1088 cm^−1^, associated with the phosphate group (PO_4_^3−^) vibrations [45]. However, compared with commercial and bovine HA, the porcine material showed less sharp bands, indicating lower crystallization [47]. The small bands at 1410 cm^−1^ and 1460 cm^−1^ in animal HA could be attributed to vibrations corresponding to the carbonate group [47]. Another significant difference between porcine- and bovine-derived HA was the peak height at the phosphate group. A more detailed emphasis on the crystallinity of HA is presented in Appendix A. The bovine peak height was comparable with the commercial and higher than porcine. The chemical composition of DBM (Figure 2f) confirmed the complete demineralization of bone, as evidenced by the absence of PO_4_^3−^ bands. Amide A bands at 3278 cm^−1^ and Amide B at 3070 cm^−1^ were found in the bovine DBM after demineralization. Similarly, amide I, II, and III bands had the highest peak intensities for bovine (Figure 2f). Comparable spectra were observed for commercial DBM and DBM extracted from porcine. The presence of amide bands I, II, and III in 1636 cm^−1^ and 1200 cm^−1^ for the extracted DBM were associated with collagen presence after extraction. The peak height for bovine-derived DBM was higher than those of bovine-derived and commercial for all of the identified peaks.

### 3.2. HA/DBM Compound Characterization

The SEM of the 50/50 compounds for both animals showed the HA particles evenly distributed along the DBM matrix (Figure 3a). Superficial measurement of the HA’s particles size showed an average of 4.57 µm ± 1.50 for the porcine compound and 7.45 µm ± 2.35 for the porcine material. Compared with the porcine HA particles, bovine showed increased agglomeration and integration with DBM. For the 60/40 compounds, the HA particles dominated the morphology due to their higher concentration in the mixture. The elemental analysis in the compounds confirmed the presence of HA with an average Ca/P ratio of 1.64 and 1.68 for the 50/50 and the 60/40 compounds, respectively (Figure 3b). There was an absence of elements associated with the chemicals used in the extraction process (i.e., HCl). The injection measurements (force−displacement tests) showed an initial linear increase in the force until the displacement reached ~2.5 mm, followed by a plateau phase at ~20 N (Figure 3c). No significant changes in the injection force were observed between the different concentrations. The results of viscoelastic tests under a fixed strain (0.5%) suggest that differences in the HA/DBM concentration did not affect the storage (G′) and loss modulus (G″) for the bovine samples. However, porcine samples showed different viscoelastic properties among the two compounds (Figure 3d). The dominance of G′ over G″ in all compounds indicates a predominantly elastic behavior. Clinical user evaluation of the compounds shows that regarding adhesion, 40% of the practitioners preferred the 50/50B compound, followed by the 60/40P (30%), 50/50P (20%), and 60/40B (10%). Regarding malleability, the 50/50 compounds had a similar behavior (35% of the users for each animal), while the less favorable was the 60/40B composition. Regarding stability, all of the compounds had similar preference values (~25%) (Figure 3e).

### 3.3. Tissue Regeneration and Inflammatory Evaluation In-Vivo

After three months of implantation, the mean calvarial tissue thickness regeneration was statistically highest in the porcine compounds, with an average of ~99% for the 50/50P, which is no different from the clinical reference (Figure 4a), followed by the 60/40P compound with ~60%. On the other hand, the lowest thickness regeneration was obtained for the bovine compounds, with only ~23% (Figure 4a). After six months of implantation, the tissue thickness regeneration had an average of 90% for all of the tested compounds (*p* ≤ 0.05) (Figure 4a). The regenerated bone area showed a similar trend to the tissue thickness regeneration after 3 months (Figure 4b). The commercial DBM and the 50/50P compound showed the highest bone regenerated area (~75%). No statistical differences were found between the 60/40P and the bovine compounds (*p* ≤ 0.05). After 6 months of implantation, all groups showed similar bone regenerated areas (~70%) (Figure 4b). The amount of regenerated collagen area after 3 months was significantly higher for the 50/50P compound and the commercial DBM (~30%) compared with the other groups (~15%) (*p* ≤ 0.05). At six months, the regenerated collagen area was statistically highest for the DBM material (~35%) (Figure 4c). All of the compounds, regardless of the animal origin, showed a similar collagen content (22%). Regarding inflammation, the blood vessel area did not differ significantly between the porcine compounds (50/50P and 60/40P) and DBM at 3 months (~2.5%). The bovine compounds had the lowest blood vessel area ~0.5% (Figure 4d); compared to three months, the area increased in all groups after 6 months. The highest blood vessel area was measured for the 60/40P and DBM (~7.0%), followed by the 50/50P compound (~5.41%). Representative histological images of naturally healed and xenograft-treated defects are presented in Figure 4e. Significant differences were observed between the treated and non-treated sites. Naturally healed defects showed isolated bone fragments (red arrows) joined by irregular collagenous tissue (green arrows) (Figure 4(eI)). Overall, the regeneration of bone was greater when defects were treated with HA/DBM compounds and the commercial DBM compared with the non-treated ones (natural healing).

## 4. Discussion

In this work, we studied the regenerative and inflammatory effects of HA/DBM xenografts extracted from porcine and bovine sources. The bone regeneration performance and blood vessel formation were evaluated in vivo using a standard rat calvarial defect [48]. Naturally healed defects without implanted grafts were used as the controls, and DBM Puros^®^ as a clinical reference. The physicochemical properties of xenografts control the bone regeneration performance, which may vary depending on the HA and DBM extraction protocols. Comparative bone regeneration studies using xenografts may use different extraction protocols; thus, the results are often contradictory [49]. The results from this work enable a comparison of the bone regeneration performance of xenografts extracted using identical protocols for HA and DBM from two animal sources. In addition, our results confirm that mixing HA and DBM from the same animal source improves tissue regeneration compared with naturally healed defects. The individual extraction and remixing of the mineral and organic phases from HA and DBM aim to overcome the challenges encountered with single-phase extractions from decellularized bone grafts, which include immunogenicity, insufficient decellularization, compromised integrity, and loss of biological activity. The individual extraction of HA and DBM allows for the customization of the mixture, improving the biocompatibility and bone regeneration potential. Controlled ratios of HA and DBM create versatile composite biomaterials suitable for bone grafts. Overall, porcine-derived compounds showed a superior regeneration performance compared with the bovine-derived compounds, but similar to the commercial DBM after 3 months of implantation.

The highest thickness regeneration (~99%) was observed in the porcine-derived mixture with the same HA and DBM content (50/50P) after 3 months. The lowest values were measured for the bovine materials with only ~25% of regeneration, regardless of the composition used. These findings are consistent with previous studies showing that porcine bone substitutes induce an appropriate cell response and have osteoconductive properties [50]. The successful incorporation of additives into HA bioceramic has been shown to facilitate adhesion, proliferation, and osteogenic differentiation of human mesenchymal stem cells. This promising outcome underscores the potential suitability of the modified bioceramic for bone implant applications [51]. Numerous factors could contribute to improved tissue regeneration. Regenerative properties of a xenograft are controlled by different graft physical properties, such as wettability [27], porosity [24], and crystallinity [52], among others [53]. For example, HA crystallinity regulates ossification and increases resorption rates during bone remodeling [52]. Go et al. demonstrated that bone particles with a low crystallinity quickly degrade and produce faster bone regeneration than high-crystalline particles [54]. Our FITR results showed the porcine-derived HA with a lower intensity, as well as broader peaks at 1023 cm^−1^ and 560 cm^−1^ associated with the movement of the PO_4_^3−^ bands (Figure 2e). Decreased intensity and broader FTIR bands are indicators of decreased crystallinity and crystal size [55]. X-ray diffraction revealed notable variations in the crystallinity of HA samples based on their origin, with porcine-derived HA exhibiting a lower crystallinity characterized by broader and less intense peaks compared with the commercial and bovine HA (Appendix A). These findings reveal that despite using identical extraction protocols, the porcine-derived HA was less crystalline than the bovine material, which could have contributed to the higher tissue regeneration performance.

Graft resorption rates are critical for the bone regeneration performance [56]. Ideally, the degradation rate of grafts should match the new bone formation rate [57]. The graft degradation rate is also controlled by material properties (i.e., chemical composition, crystallinity, particle size, surface area, and porosity) and local biological conditions (i.e., pH, temperature, and H_2_O content) [56]. Previous studies have shown that the resorption of HA/DBM grafts is delayed when the mineral quantity is higher than the organic, regardless of whether crystalline or non-crystalline HA is used [58]. In our study, in vitro biodegradation profiles showed significant differences in the degradation rates between different animal species and concentrations, where the degradation rate in porcine was found to be higher than that in bovine (Appendix A). In vivo results, after 3 months of implantation, compounds with a higher HA concentration, displayed less regeneration (~60% regeneration for porcine compounds vs. ~23% for the bovine materials), which may be associated with lower resorption rates (Figure 4). Previous studies have shown that deproteinized bovine bone grafts have a slow rate of degradation, as after 3 years of implantation, most of the graft remain intact [59]. Our results agree with these observations as compounds with a lower quantity of DBM showed an improved bone regeneration performance. The structural composition of collagen in DBM can also influence the bone regeneration performance [60]. FTIR bands of collagen corresponding to amide A, amide B, amide I, amide II, and amide III peaks [61] have been associated with crosslinking and changes in the amide functional groups [52]. Our results showed that after extraction, bovine DBM had higher intensity peaks at 1541 cm^−1^ compared with porcine (Figure 2), indicating an increase in the number of amide bonds related to increased crosslink density. High-intensity peaks in the amide structures are associated with a tight pack of collagen molecules with delayed graft resorption [52]; meanwhile, decreased intensity indicates fast collagenase hydrolysis, digestion, and resorption [62]. The increased crystallinity in our bovine-derived DBM could also contribute to a slower regeneration performance.

Bone vasculature is essential for many processes, such as skeletal development and growth, bone modeling, and remodeling [63]. Most inflammatory components transit through the blood and vessels by providing the necessary nutrients for healing, while eliminating waste products from the bone grafts [64]. A fast degradation rate of bone grafts may elicit an inflammatory response due to graft debris accumulation [62]. The number of blood vessels and bone regeneration are closely related processes [65]; increased angiogenesis improves bone healing [66]. For example, Lu et al. showed that a decreased vascular perfusion in bone reduced callus size, lowered cell proliferation, and increased apoptosis. This resulted in fibrous and adipose tissue formation over cartilage and bone [67]. In agreement with this observation, after 3 months of implantation, the compounds with the highest formation of blood vessels (i.e., DBM, 50/50P, and 60/40P) showed the highest tissue regeneration (Figure 4). In contrast, the bovine substitutes had the lowest tissue regeneration and blood vessel formation. At 6 months, we did not find a relationship between the percentage of blood vessels and the regenerated tissue area. Similar to our findings, previous reports suggest a stronger relationship between early periods of blood vessel formation and early bone formation [68]. As inflammation and vascularization have a characteristic timeline directly associated with the first few days and months after the injury, 6 months is considered a late stage of maturity. Therefore, no associations were expected [69]. In addition to histology tests, IL-6 and bone expression proteins (*RUNX2*, *COLI*, and *ALP*) were also assessed and found to be increased in porcine grafts, as confirmed by PCR. Early responses were observed at 7 days for the expression of *RUNX2, COLI,* and *ALP*, indicating active bone remodeling and differentiation processes at the graft site (Appendix A). The combined results from histology and PCR suggest that porcine grafts exhibited a robust and accelerated bone healing response compared with other graft types. M. Li cited that pro-inflammatory genes, such as TNF-α, could be down-regulated by decreasing the Ca^2+^ concentration, while pro-healing genes including IL-6 were simultaneously up-regulated [70].

A common practice in bone regeneration studies is to use commercial products as a reference material for proper benchmarking [67]. Most commercial grafts are comprised of DBM only. Previous studies have shown that when DBM-only xenografts are used for large-sized defects, the regenerated tissues are collagenous, dominated by low-quality bone formation [71]. Regenerated tissues with these qualities have been related to tissue collapse after implantation [72]. Our results are consistent with these observations. The commercial DBM used in this study showed a statistically higher collagen content after 6 months of implantation (~35%) compared with the HA/DBM compounds (~25%) (Figure 4c). The addition of a mineral material (i.e., HA) to the xenograft induces extracellular matrix mineralization and the formation of apatite layers similar to bone. As a result, less collagenous tissues are regenerated and, thus, there is a higher bone quality [73]. Xenografts can trigger osteogenesis that extends over the native calvarial plateau [74]. Our findings suggest that filling large-sized defects without xenografts favors fibrotic scar tissue (Figure 4e). These observations were consistent with a previous study [75]. The implanted materials are directly associated with the healing process.

Despite the significant results, this study still has limitations. First, although we used a similar extraction protocol to properly compare differences between the two species, our extraction processes were not fully optimized for bone regeneration performance. Optimization of the extraction protocol could render improved regeneration. Second, the regeneration achieved after 3 and 6 months of implantation was considered to be “late regeneration” [76]. An early-stage (1 month) regeneration process using imaging techniques, including micro-CT [77] and Masson’s trichrome stain [78], could provide a better understanding of bone regeneration and the inflammatory response of these animal-extracted xenografts.

## 5. Conclusions

In this study, for the first time, we evaluated the bone regeneration and inflammatory effects of xenografts obtained by mixing HA and DBM extracted from the same animal sources (i.e., porcine and bovine). All animal-derived xenografts provided improved tissue regeneration compared with the naturally healed defects. After 3 months of implantation, porcine-derived HA/DBM compounds showed improved tissue regeneration compared with the bovine-derived ones. Specifically, the tissue thickness regeneration was 50% higher than in the bovine-derived materials. The tissue regeneration was comparable to a clinical reference (DBM-only). HA/DBM compounds showed less regenerated collagen compared with DBM-only xenografts. No clinical complications associated with any implanted compound were found during the study. The differences found in the three-month regeneration appeared to be related to the reduced crystallinity and cross-linking of the porcine-derived HA and DBM, respectively.

## Figures and Tables

**Figure 1 jfb-14-00439-f001:**
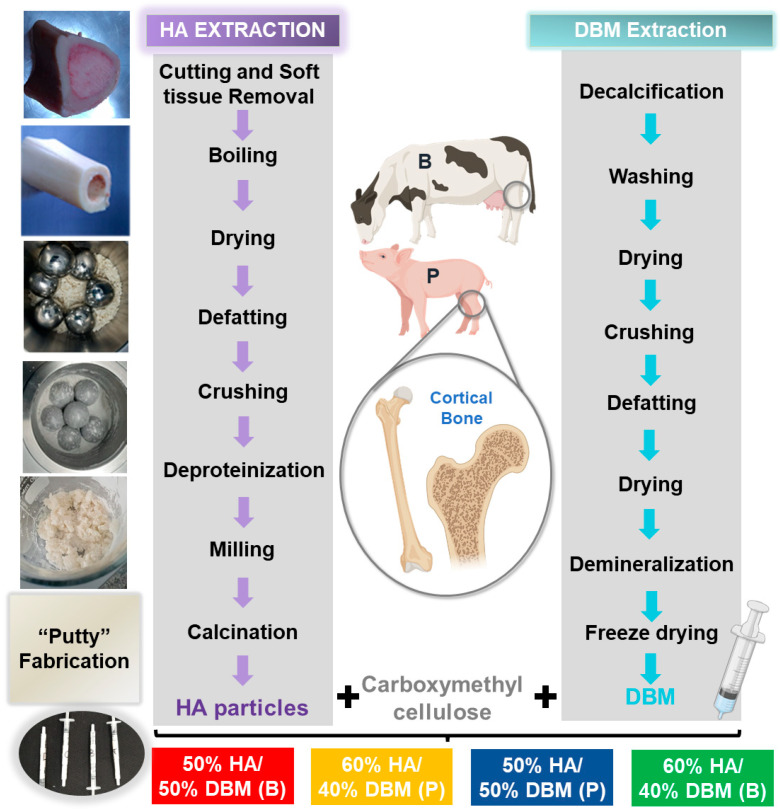
Schematic representation of the extraction process of hydroxyapatite (HA) and demineralized bone matrix (DBM) from bovine (B) and porcine (P) sources.

**Figure 2 jfb-14-00439-f002:**
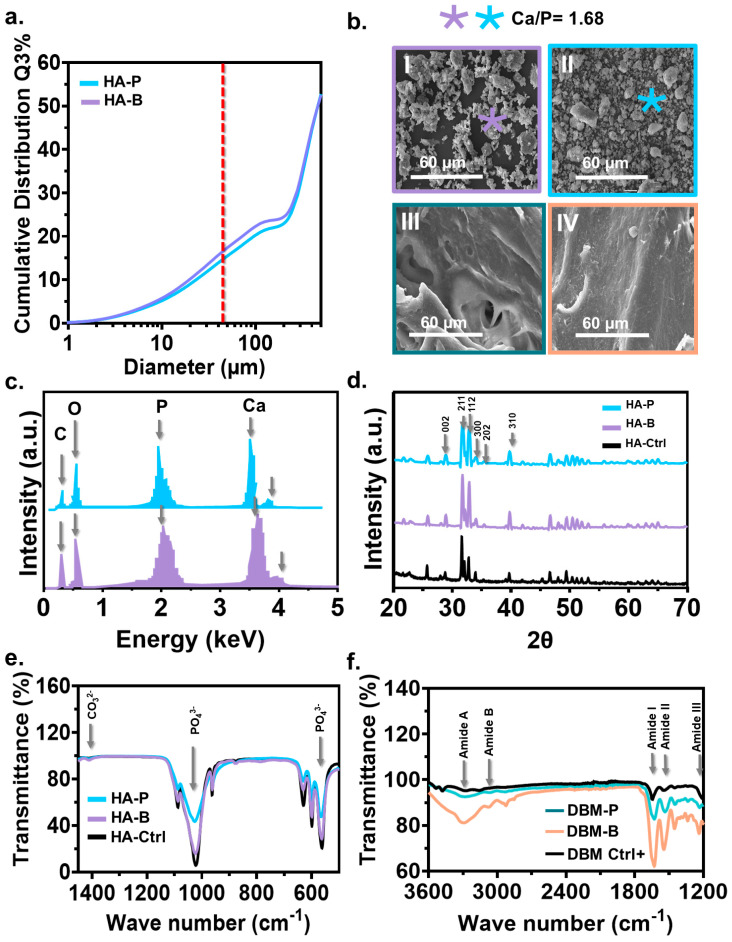
The characterization of HA and DBM from bovine and porcine sources. (**a**) Particle size distribution of the HA extracted from porcine (P) and bovine (B) sources. Only particles smaller than 45 mm (red line) were considered for compound preparation. (**b**) Representative SEM micrograph of the extracted HA and DBM. (**c**) EDS analysis obtained from a randomly selected region (marked spot) of the extracted HA and DBM. (**d**) X-ray diffraction pattern of the extracted HA and their lattice planes. (**e**) Selected FTIR spectra of the extracted HA indicate the characteristic peaks corresponding to the vibrations of the phosphate (PO_4_^3−^) groups. (**f**) The selected FTIR spectra of the extracted DBM indicate the characteristic bands associated with amide groups.

**Figure 3 jfb-14-00439-f003:**
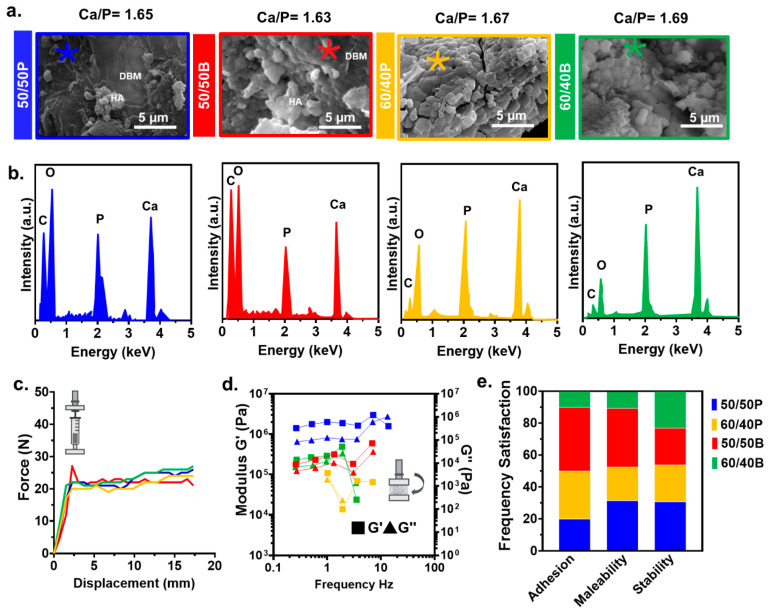
Characterization of the HA/DBM compounds. (**a**) Representative SEM micrographs of 50/50P (blue), 50/50B (red), 60/40P (yellow), and 60/40B (green) compounds. Regions in the SEM image where elements were analyzed are denoted by an asterisk (*). (**b**) The representative EDS spectrum of the HA/DBM compounds showed peaks corresponding to the materials present in the mixtures. (**c**) Injection force required for the compounds to be injected through a 27-gauge needle. All of the measurements were made, simulating a flow rate of 2 mL/h. (**d**) Frequency dependence of the storage modulus (G′) and loss modulus (G″) in HA/DBM compounds measured at 0.5% strain. (**e**) Clinical user preference according to their adhesion, malleability, and stability.

**Figure 4 jfb-14-00439-f004:**
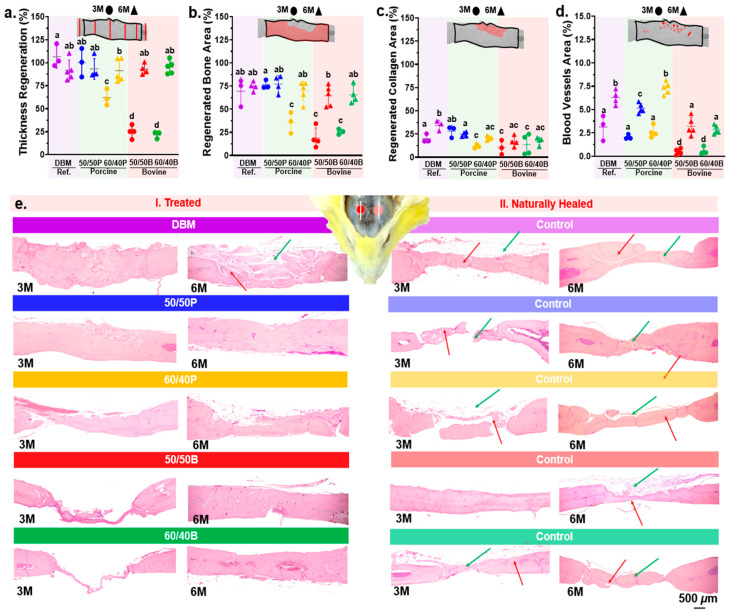
Histological and histomorphometric analysis of the regenerated tissue in calvarial bone defects after the implantation of HA/DBM compounds. Results for 3 and 6 months after implantation are shown using circles and triangles, respectively. Colors represent each of the groups. (**a**) Percentage of calvarial thickness regeneration. (**b**) Percentage of regenerated bone. (**c**) Percentage of regenerated collagen. (**d**) Percentage of blood vessel area with respect to the total regenerated area. (**e**) Representative histological images of the treated defects (**I. Left images**) and naturally healed (**II. Right images**). Bone fragments and collagen tissue are indicated with red and green arrows, respectively. Means with different letters represent values that are significantly different from each other (*p* ≤ 0.05).

## Data Availability

Research data supporting this publication are available. All relevant data are within the paper and its Appendix A.

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
