# Peer review of "A Comparative Study of HA/DBM Compounds Derived from Bovine and Porcine for Bone Regeneration"

_jfb, 2023, doi:10.3390/jfb14090439_

Round 1
Reviewer 2 Report
This is a article where authors assessed for the first time the bone regeneration and inflammatory consequences of xenografts made by combining HA and DBM taken from the same animals. it was well written in all its parts with fluent and clear English. The schematization in several paragraphs helped to understand the complexity of this study in a simple and effective way.
Kind Regards
Reviewer 3 Report
The authors extracted the HA and DBM contents of bovine and porcine bone and mixed them in different portions to evaluate their performance in terms and inflammatory responses and bone regeneration. The study was clearly designed and the results and discussion were well-prepared. However, a few comments need to be addressed before acceptance:
1- Why did the authors extract both the mineral and organic phases, when they wanted to mix them again? The bones could be easily decellularized to obtain the final mixture.
2- The highest thickness regeneration in the porcine-derived mixture was attributed to the crystallinity of HA parts, It would be informative to analyze and compare the crystallinity of the HA component of the commercialized control grafts to confirm the discussion.
3- Evaluation and comparison of the in vitro biodegradation profiles are recommended.
4- The cited references sound too much (100 refs). It looks like a review article rather than a research.
Reviewer 4 Report
Comments file:
The manuscript titled “A Comparative Study of HA/DBM Compounds Derived from Bovine And Porcine For Bone Regeneration” is written well and discusses a very important area of research that has clinical significance. However, there are a few things that need to be considered before the paper can be accepted for publication.
1. The duration of the study and the study design needs to be explained in detail.
2. References need to be updated. Newer citations and more relevant ones are needed in place of the older citations.
3. The authors need to scrutinize the document and make changes to the grammatical errors as well as human errors.
4. Inflammation and the subsequent steps are needed for adequate regeneration. The macrophages in the tissue turn into osteoclasts and help in the regeneration process. Additional data with IHC stains for CD45 markers might be helpful.
Round 2
Reviewer 1 Report
The revised version of the manuscript could be considered for publication.
The revised files of the manuscript could be considered for publication.